# Planktic foraminifera form their shells via metastable carbonate phases

D.E. Jacob [1], R. Wirth[2], O.B.A. Agbaje[1], O. Branson [3] & S.M. Eggins[3]

The calcium carbonate shells of planktic foraminifera provide our most valuable geochemical archive of ocean surface conditions and climate spanning the last 100 million years, and play an important role in the ocean carbon cycle. These shells are preserved in marine sediments as calcite, the stable polymorph of calcium carbonate. Here, we show that shells of living planktic foraminifers *Orbulina universa* and *Neogloboquadrina dutertrei* originally form from the unstable calcium carbonate polymorph vaterite, implying a non-classical crystallisation pathway involving metastable phases that transform ultimately to calcite. The current understanding of how planktic foraminifer shells record climate, and how they will fare in a future high-$CO_2$ world is underpinned by analogy to the precipitation and dissolution of inorganic calcite. Our findings require a re-evaluation of this paradigm to consider the formation and transformation of metastable phases, which could exert an influence on the geochemistry and solubility of the biomineral calcite.

[1] Department of Earth and Planetary Sciences, Macquarie University, Sydney, 2109 NSW, Australia. [2] Helmholtz-Centre Potsdam, German Research Centre For Geosciences GFZ, 14473 Potsdam, Germany. [3] Research School of Earth Sciences, Australian National University, Canberra, 2601 ACT, Australia. Correspondence and requests for materials should be addressed to D.E.J. (email: Dorrit.jacob@mq.edu.au)

Planktic foraminifera are among the most important calcifying organisms in the open ocean, contributing as much as half the particulate CaCO₃ exported from the surface ocean annually (ca. 2.9 Gt CaCO₃ yr⁻¹)[1, 2]. Their calcite shells, preserved in the marine sedimentary record over the last 500 million years[3], provide an unparalleled geochemical archive of past climate in their trace element and isotope chemistry. Translating this archive into useful climatic information requires a systematic understanding of how mineral growth conditions relate to trace element chemistry, historically established by analogy to synthetically produced calcite[4]. However, the geochemistry of foraminiferal shells diverges significantly from inorganic calcite precipitated from seawater-like solutions[5]. These differences have been attributed to vital effects[6, 7], which encompass all the influences that biological processes might exert on foraminiferal calcite composition[5, 8–10].

All existing models of foraminiferal mineralisation assume shell formation proceeds via the direct precipitation of calcite. However, many biologically precipitated carbonates are known to form via complex mineralisation pathways involving metastable intermediate phases that transform stepwise into the final shell mineral[11]. This provides an energetically and kinetically favourable pathway to calcite formation, by employing metastable particles with high surface energy, which have a lower free energy barrier to nucleation[12]. If similar processes are employed by foraminifera, they would provide an alternate mineralogical control on shell geochemistry, and could account for the discrepancies between the chemistry of foraminifera shells and inorganic calcite precipitated from seawater-like solutions. This would alleviate the requirement for complex, energy intensive ion control mechanisms to explain the geochemistry of foraminifera.

*Orbulina universa* (Fig. 1a) and *Neogloboquadrina dutertrei* (Fig. 1b) are two common, well-studied species of planktic foraminifera that are frequently employed in palaeoceanographic[13] and biomineralisation studies[14–16]. *O. universa* is representative of spinose and *N. dutertrei* of non-spinose groups of symbiotic planktic foraminifera. Both construct their shells by the sequential addition of distinct 'chambers'[17], which are formed of calcium carbonate with a very low-Mg content (0–10 mmol/mol Mg/Ca[18]). The formation of each chamber begins with the extrusion of a thin, balloon-like organic template in the form of the new chamber. Previous TEM investigations reveal micron-scale plaques of CaCO₃ develop and coalesce to form continuous layers of CaCO₃ on both sides of the organic template[19, 20], resulting in a characteristic bilamellar chamber wall construction. The architecture of the shell walls of both species is highly organised, and contains numerous pore channels through which metabolic substrates and products are transported (Fig. 1c, d). *O. universa* also supports long CaCO₃ spines that radiate up to several hundred micrometres from the shell surface and play important functions in foraminiferal biology (Fig. 1a). *O. universa* is unusual amongst foraminifera in that its final mineralised chamber is large (400–1000 μm diameter) and fully encloses the earlier formed chambers. The formation of this final chamber is likely analogous to chamber wall thickening in other species of foraminifera[15].

We present an in situ study of the ultrastructure and mineralogy of *O. universa* and *N. dutertrei*, using focussed ion beam (FIB) supported transmission electron microscopy (TEM) imaging and diffraction, and Fourier transform infrared (FTIR) spectrometry.

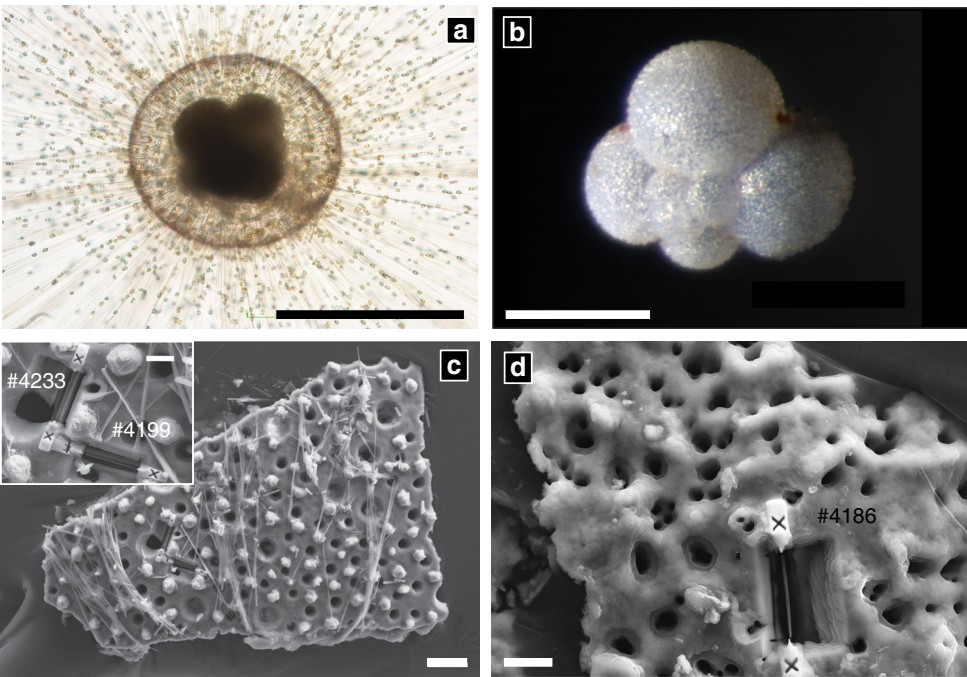

**Fig. 1** The structure of foraminiferal shells. **a** *Orbulina universa* develops a final, large, spherical chamber, which encloses the earlier formed spiral shell of its 'juvenile' stage (dark, centre). The shell supports long radiating mineralised spines that host a sticky network of streaming rhizopodia that are used to trap prey, take up and excrete material, and deploy algal symbionts. This final spherical chamber is initially thin-walled and thickens continuously over 3 to 7 days. Near the end of the foraminifer's life cycle[47], the spines are shed and *O. universa* undergoes a massive release of gametes, leaving an empty shell that sinks rapidly to the sea floor, exporting CaCO₃ from the surface ocean. **b** *Neogloboquadrina dutertrei* form trochospiral shells of consecutively larger chambers, and do not have mineralised spines or symbionts. Electron-transparent foils were extracted orthogonally from the outer surface (Session 1) or polished cross-sections (Session 2) of pre-gametogenic *O. universa* (**c**) and *N. dutertrei* (**d**) shell fragments using a FIB. The pits in the images show the locations of foil extraction with foil identification numbers. Scale bars are 500 μm (**a**), 200 μm (**b**), 20 μm (**c**), 5 μm (inset), 5 μm (**d**)

**Table 1 Details on storage and measurements**

| Analytical session | Storage | | Technique | Preparation |
|---|---|---|---|---|
| | Months | Conditions | | |
| 1 | 20 | Dry | TEM | Surficial FIB extraction after shell fracture. |
| 2 | 26 | Dry | TEM | Cross-section FIB extraction after resin-mount and polish. |
| 3 | 39 | Dry | FTIR | Unprepared, analysed broken. |
| 4[a] | 1 | Wet | XRD | Shells broken open by hand. |

TEM transmission electron microscopy, FTIR Fourier transform infrared spectroscopy, XRD X-ray diffraction, FIB focussed ion beam
The storage time, conditions, measurements conducted and sample preparation for specimens measured in all four analytical sessions. Specimens measured during sessions 1–3 were all collected at the same time, and stored in the same conditions
[a]Separate collection of *N. dutertrei* only

## Results

**Sampling.** Shells of *O. universa* (Fig. 1c) and *N. dutertrei* (Fig. 1d) were live-collected and analysed using a range of techniques over four analytical sessions after varying time spent stored under wet or dry conditions (Table 1).

**Shell ultrastructure.** Shell ultrastructure was examined by high-resolution TEM (HR-TEM) during our first two analytical sessions (Table 1). Electron-transparent foils (ca. 150 nm thick) were extracted from shells of *O. universa* (Fig. 1c) and *N. dutertrei* (Fig. 1d) by FIB milling. For session 1, foils were cut orthogonal to the outer shell surface (Fig. 1c, d), while for session 2 they were extracted from polished shell cross-sections. All TEM analyses revealed a microstructure of fibrous structures (Fig. 2a) with abundant, nanometre-sized pores (distinct from the larger structural pore channels that bisect the shell wall; Fig. 2c, Supplementary Fig. 2). The fibrous structures are aligned orthogonal to the shell surface, and show similar diffraction contrast (Supplementary Fig. 4a), indicating a consistent crystallographic orientation. They are ~40 nm in diameter in *O. universa* (Fig. 2b) and 100 nm in *N. dutertrei* (Supplementary Fig. 3), with undulating, interlocking margins. These characteristics are typical of crystals formed via particle attachment[11, 21], and the nanopores in these crystals may be a consequence of imperfect packing and volumetric changes during particulate assembly and transformation[11]. These nanopores are similar to the abundant voids seen in crystals formed via particle-attachment in vitro, although they are likely filled with organic material or water in natural biominerals and these foraminifer shells[22]. The tips of the fibres form a ragged plane of oriented single crystals at the shell's outer surface (Supplementary Fig. 4b, c) with crystallographic *c*-axes aligned radially along the shell growth axis, and are protected from the external environment by a ca. 30 nm-thick organic layer (Fig. 2a inset, ref. [20]).

**Shell crystallography and infrared spectrometry.** Thirty electron diffraction patterns were collected from both foraminifer species during our first analytical session. All are consistent with the structure of vaterite, rather than calcite, with hexagonal unit cell parameters $a_0 = b_0 = 0.71239$ nm; $c_0 = 2.53204$ nm[23] (Supplementary Table 1, Supplementary Fig. 1). Vaterite has a complex, layered crystallographic structure, where variations in symmetry between the layers create a number of polytypic structures[24, 25]. The crystallographic structure of vaterite we observe in these foraminifer shells is one of the most ordered variants and has a six-layer periodicity[24, 25].

Electron diffraction patterns collected from the entire foils during session 1 (Fig. 3a) reveal a prominent lattice spacing of 0.841 nm (Fig. 3, Supplementary Table 1). This lattice spacing fits neither calcite nor vaterite but rather corresponds to a twinning superstructure, in which vaterite crystals (Fig. 3b–e) in neighbouring fibres in the shell are twinned by 180° rotation around a

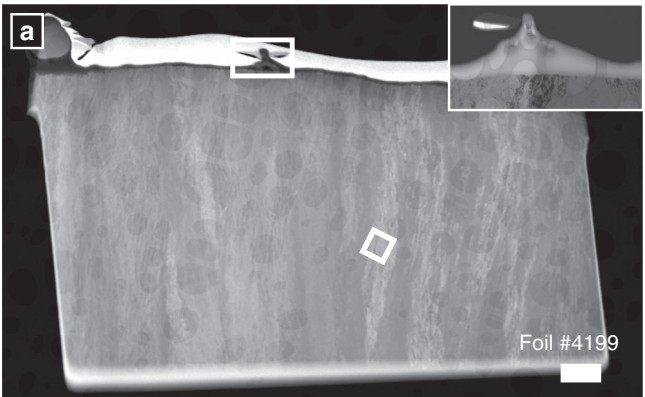
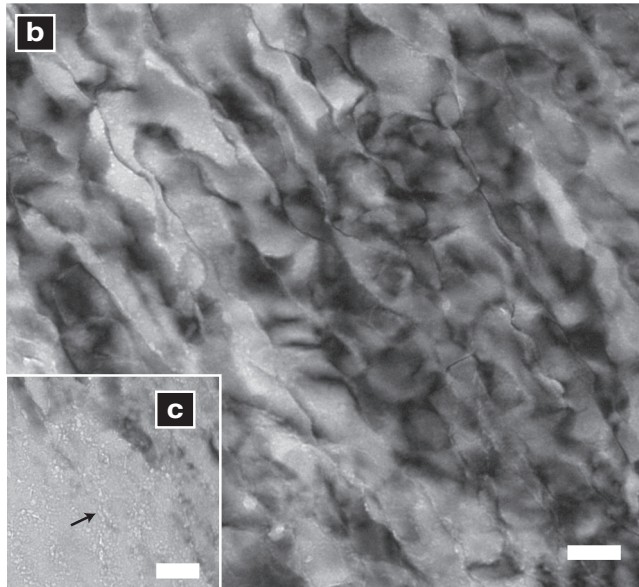

**Fig. 2** Microstructure of *O. universa* shells. **a** High angle annular dark field (HAADF) image shows the fibrous texture of the chamber and the base of a spine (white rectangle and enlarged in the inset). Note the organic membrane that covers the spine foundation in the inset. The circular features in the sample image are caused by the underlying carbon sample holder. **b** Bright-field image of the particulate fibres that comprise the chamber wall. **c** area in foil #4233 showing pores (arrow). Scale bars are 1 μm (**a**), 50 nm (**b**) 50 nm (**c**). Analyses were carried out during session 1

shared crystallographic *c*-axis (aligned orthogonal to the shell surface). This twinning structure may enhance the physical properties of the foraminifer shell by averaging out the mechanical anisotropy of the crystals and optimising the mechanical stability of the shell, as has been observed for Dauphiné twins in quartz[26].

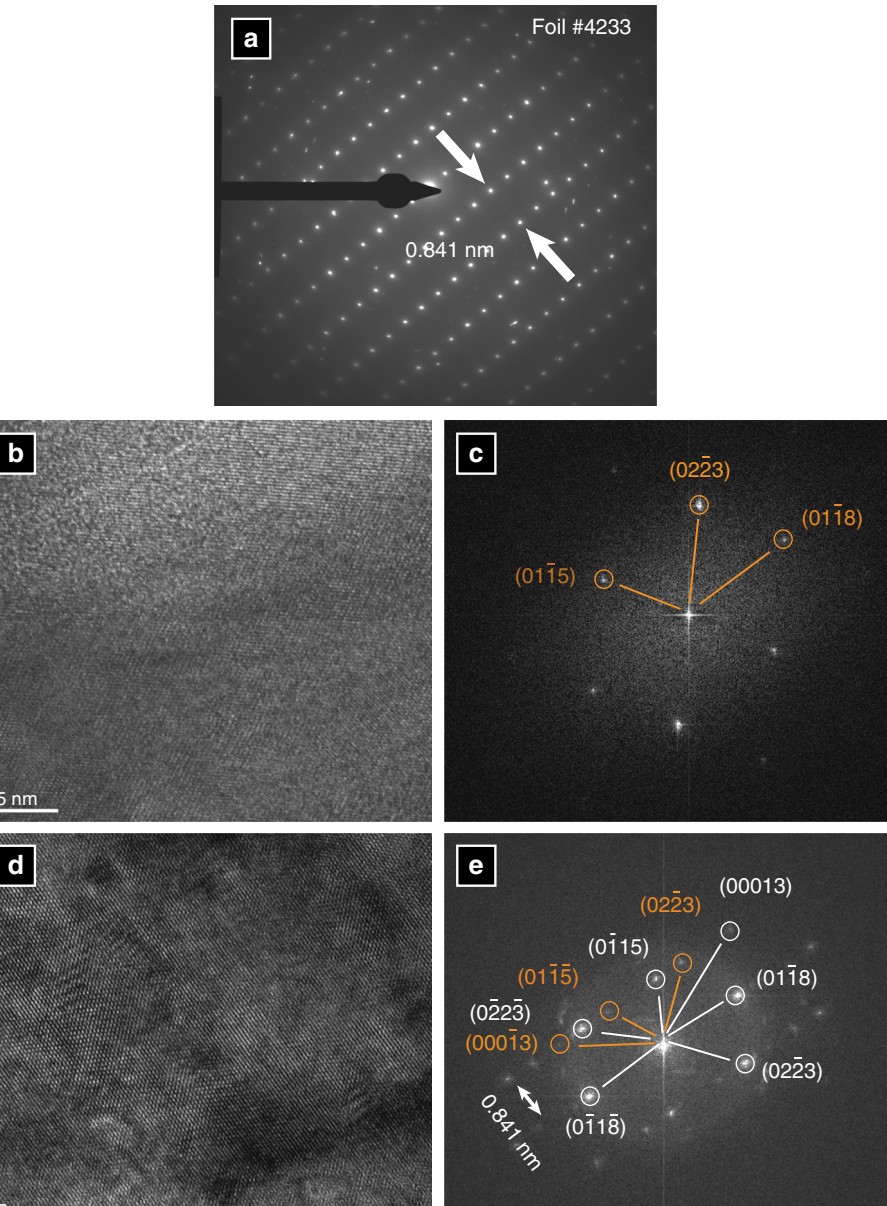

**Fig. 3** Twinned structure of *O. universa* shells. Electron diffraction patterns collected from an entire TEM foil (**a**) show a prominent lattice spacing of 0.841 nm, which is consistent with neither calcite nor vaterite. High-resolution analysis of a crystallographically uniform region within the foil (**b**) via Fourier Analysis revealed a characteristic vaterite electron diffraction pattern (**c**; orange). Similar analysis of a crystallographically complex region within the superstructure (**d**) revealed that the 0.841 nm lattice spacing observed in the entire foil can be explained by a twinned vaterite structure, with twin pairs rotated 180 along the *c*-axis (**e**; orange vs. white). All analyses carried out in analytical session 1

Session 1 HR-TEM analyses further revealed that not all areas within the chamber wall sections were crystalline. Fourier transform analyses of HR-TEM images of the rim of the vaterite fibres (Fig. 2b) display diffuse 'Debye-Sherrer rings' (Fig. 4a, b), and lack lattice fringes observed in crystalline materials (Fig. 4c, d). These are characteristic of non-crystalline materials that could either be regions of amorphous calcium carbonate (ACC), or organic components included within the shell structure[16].

Eighteen additional electron diffraction patterns were collected during analytical session 2. All are consistent with the structure of calcite, in keeping with the long-standing paradigm of foraminiferal mineralogy[18], and contradicting the results from our first analytical session. However, vaterite is highly unstable and its absence in our second set of analyses does not negate our initial results. It is likely that differences in sample treatment caused a

vaterite-calcite phase transformation prior to analysis. The only differences between the analytical sessions were an additional 6 months of storage, prior mounting in resin and the polishing of specimens before FIB extraction for session 2; FIB-milling procedures and TEM analytical conditions were identical. An additional 6 months of storage under similar conditions is unlikely to have induced a phase transformation. Sample mounting and polishing exposes the sample to thermal energy during exothermic resin hardening, and mechanical energy in a hydrous environment during polishing. Any of these processes could have provided the necessary energy and environment to facilitate a phase transformation. Two additional analyses were conducted to test the veracity of our initial vaterite result, and explore possible conditions that lead to a phase transformation: FTIR and X-ray diffraction (XRD).

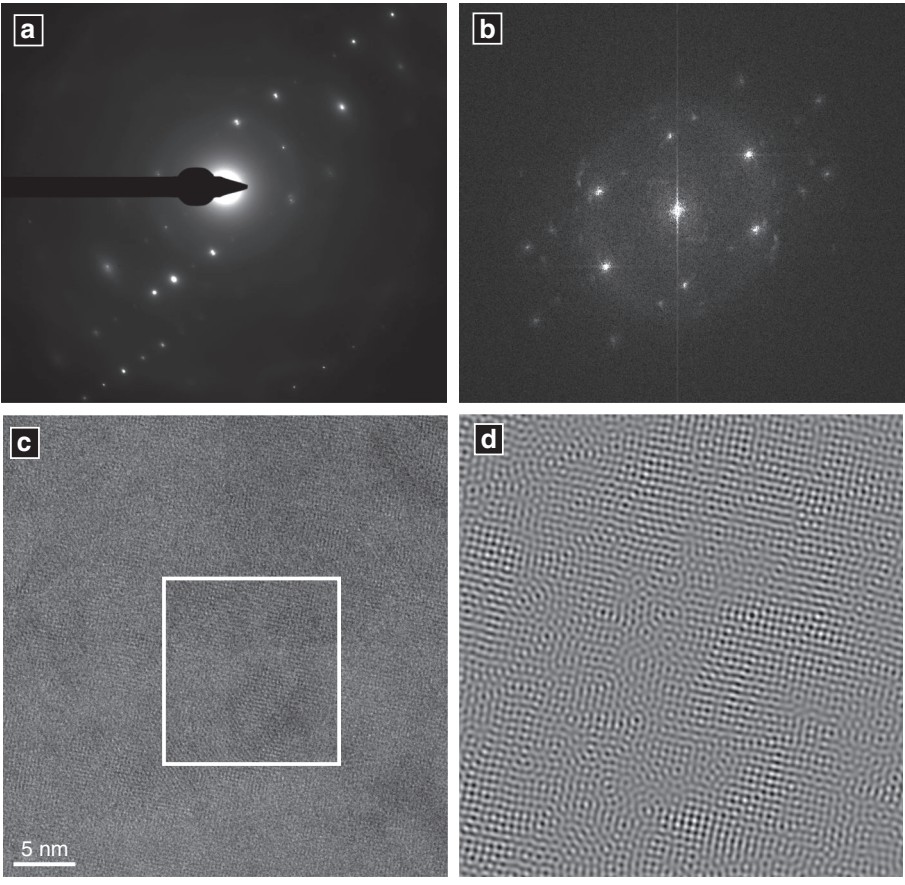

**Fig. 4** Amorphous regions on crystal fringes. Electron diffraction patterns collected from *O. universa* during analytical session 1 (**a** image plate detector, **b** Fourier Transform analysis of (**c**)) display Debye-Scherrer rings (diffuse diffraction patterns around the image centre). These are typical of amorphous material. After filtering and inverse Fourier transformation (**d**) areas devoid of lattice fringes can be seen that are inconsistent with a crystalline material, but consistent with the presence of either ACC or organic material. Lattice fringes are visible as periodic parallel lines in **c** and **d**

Two individual shells each of *O. universa* and *N. dutertrei* from the same sample batch measured during session 1 and 2 were measured by attenuated total reflection FTIR (ATR-FTIR; Session 3), and compared to spectra collected from vateritic spicules of *Herdmania momus*[27] and geological calcite (Fig. 5, Supplementary Fig. 5). Vaterite is identified in FTIR spectra by a characteristic shift in the $\nu_4$ $CO_3^{2-}$ vibration band from ~712 cm$^{-1}$ in calcite, to ~744 cm$^{-1}$, caused by changes in the local bonding environment of the $CO_3^{2-}$ group[28]. Based on the relative areas of these diagnostic peaks[29], *O. universa* contained ~4.5% vaterite and *N. dutertrei* contained ~3%. These analyses independently confirm the presence of vaterite in both species. Furthermore, these analyses were conducted on specimens from the same batch as TEM analyses during our first two sessions, indicating that the lack of vaterite in session 2 is most likely attributable to phase transformation during sample preparation, rather than storage conditions. The relatively small percentage of vaterite present in these specimens is at odds with the entirely vaterite composition of FIB foils observed during analytical session 1. This indicates a gradual transformation of vaterite to calcite during storage, leading to a decrease in amount of observable vaterite with time.

To further explore the effect of sample storage on foraminiferal vaterite stability, we analysed newly-collected *N. dutertrei* specimens by XRD in analytical session 4. These specimens were stored in wet saline conditions for 1 month, and contained no detectable vaterite on analysis. This preliminary result suggests that a hydrous environment may be important in facilitating a

vaterite-calcite phase transformation in foraminifera, and that rinsing in pure water and storage in dry conditions may be responsible for vaterite preservation in our other specimens. A more rigorous study of sample treatment and storage conditions is required to fully evaluate this.

### Discussion
Our HR-TEM analyses show that the planktic foraminifers *O. universa* and *N. dutertrei* mineralise their calcite shells via vaterite. The presence of vaterite in minimally prepared, dry-stored samples and its absence in samples that were either energetically prepared or stored in a wet, saline environment suggests metastable vaterite is an important early phase to form in the organism that subsequently transforms to stable calcite in the natural environment. Thus, these foraminifera employ a non-classical crystallisation pathway that involves the transformation of metastable vaterite into calcite. The presence of vaterite in both species strongly suggests that other foraminifera, which also produce low-Mg calcite shell composition, likely share a similar biomineralization strategy. The highly unstable nature of vaterite has likely contributed to its occurrence in foraminifer shells remaining undiscovered until now. Our findings supersede the long-standing paradigm that planktic foraminifera construct their shells by the direct precipitation of calcite, which has persisted since XRD analysis was first applied to foraminifer shells[18]. Our findings add planktic foraminifera to the growing catalogue of calcifying taxa, including sea urchins, molluscs, sponges and

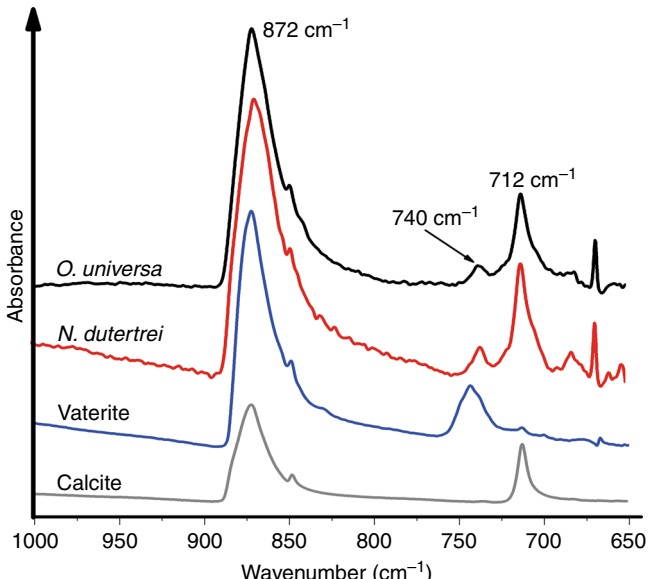

**Fig. 5** Fourier transform infrared spectra of geological calcite and vateritic *Herdmania momus* spicules compared to *O. universa* and *N. dutertrei* shells. The absorption bands at ~740 and 712 cm$^{-1}$ are the $\nu_4$ vibrational frequencies of in-plane bending of the carbonate ion[28]. The carbonate ion in vaterite has a higher vibrational frequency (~740 cm$^{-1}$) than in calcite (712 cm$^{-1}$) because of differences in the bonding environment of the ion, and these bands are diagnostic of the two mineral phases. The ~740 cm$^{-1}$ peak in the foraminiferal specimen is slightly offset from the vaterite peak in *Herdmania momus* (~743 cm$^{-1}$). These peaks are the same within spectral resolution of the instrument (4 cm$^{-1}$), although this offset could also indicate subtle differences in vaterite structure. The relative areas of these bands suggest that the foraminifera contain ca. 4.5% vaterite in *O. universa* and 3% in *N. dutertrei* (ref. [29], see Methods). All analyses carried out in analytical session 3

crustaceans[30, 31], that employ metastable intermediate phases to form their shells and skeletons[32, 33].

The mechanisms by which vaterite transforms to calcite are unstudied, but a considerable body of work has explored non-classical crystal growth involving the transformation of metastable ACC. In vitro transformation of ACC to crystalline calcium carbonate proceeds via dissolution and reprecipitation[34–36], whereas most in vivo studies of skeletal structures reveal a solid-state process involving dehydration and structural re-arrangement[32, 37]. A combination of dissolution-reprecipitation and solid-state transformation processes is possible, with the predominance of either being influenced by the amount of aqueous fluid present[29, 34]. The similarity of the twinned, fibrous microstructure and crystallographic orientation observed in both vateritic (Session 1) and post-transformation calcitic (Session 2) foraminifer shells suggests either a solid-state mechanism, or a highly-localised dissolution-reprecipitation process at the vaterite-calcite boundary[29].

Solid-state transformation from ACC to calcite is common in vivo[30, 34], but significant crystallographic differences between vaterite and calcite create a high-energy barrier that makes solid-state transformation between these two phases much less likely. The absence of vaterite in wet-stored specimens (Session 4) suggests a dissolution-reprecipitation transformation facilitated by hydrous environments is most likely. Furthermore, the large calcite single crystal observed bisecting the calcite shell micro-structure of a dry-stored *N. dutertrei* sample in analytical session 2 (Fig. 6) is unlikely to be the product of a solid-state transformation, which is more likely to preserve the micro-crystalline

architecture of the vaterite. Together, these observations indicate a dissolution-precipitation transformation mechanism is most likely, similar to that observed in in vitro ACC experiments[34]. Our results do not constrain the exact mechanism or timing of vaterite-calcite transformation, but do indicate that living foraminifera form their calcitic shells via vaterite. It further is possible that vaterite is preceded by another less-stable phase such as ACC, although this has not yet been demonstrated for foraminifera. This has significant implications for our understanding of geochemical proxies[36], and potentially the vulnerability of foraminifera to a future, more acidic ocean.

Our results require a re-evaluation of the incorporation mechanisms of trace elements and isotopes into foraminiferal calcite, to account for the influence of precipitation of a meta-stable vaterite phase and possible initial ACC phase on shell geochemistry. This is not feasible at present because we lack knowledge of both vaterite geochemistry and the nature of the vaterite-calcite phase transformation. The only existing data that provides insight into the possible influence of vaterite on for-aminiferal calcite composition is the fractionation of calcium isotopes during precipitation of synthetic vaterite[38].

Vaterite, calcite and *O. universa* shells are all enriched in $^{40}$Ca relative to the fluid they precipitate from[38–40]. *O. universa* shells are most enriched and vaterite is least enriched in $^{40}$Ca. This also rules out a solid-state transformation that transfers all the Ca in vaterite to the resulting calcite, which would result in *O. universa* inheriting the Ca-isotope composition of vaterite. Rather, it implies foraminiferal calcite forms via a process that involves multiple fluid-mineral fractionation steps. A vaterite to calcite dissolution-reprecipitation pathway could provide a double-fractionation mechanism, the first on formation of vaterite and a second on transformation to calcite in the presence of a solution phase that is not isolated from the external environment (Fig. 7). This multi-step process is qualitatively consistent with the $^{40}$Ca enrichment of calcite, which is of similar magnitude to the difference between vaterite and *O. universa*[38–40].

A multi-step crystallisation pathway involving vaterite, and possibly ACC, would have a significant influence on all aspects of the trace element and isotope geochemistry of foraminiferal calcite (Fig. 7). The current lack of vaterite and ACC geochemical data prevent us from quantifying this influence, but a double-fractionation process provides a conceptual framework that might account for the significant and unexplained differences between foraminiferal and inorganic calcite geochemistry. Most notable is the order of magnitude lower Mg content of foraminifera (0.1–1.0 mol%) compared to inorganic calcite (10–20 mol%; [4]). This has attracted the proposal of complex, often contradictory mechanisms involving energetically expensive selective pumping of Mg or Ca ions out of or in to the calcification environment[5, 8–10], and has hindered understanding of the widely used foraminiferal shell Mg/Ca proxy for past ocean temperature[41]. If Mg partitioning into vaterite is of similar magnitude to calcite, a double-fractionation provides a straightforward explanation for the low-Mg content of foraminiferal calcite. New experiments investigating the geochemistry of vaterite formation and the vaterite-calcite phase transformation stand to transform our understanding of foraminiferal shell geochemistry.

The presence of vaterite as an intermediate mineral phase in foraminiferal shells may make them more susceptible to dissolution in a future, more acidic ocean, since vaterite is more soluble than calcite[1]. While living foraminifera possess a protective organic layer on the shell surface the degradation of this membrane after death will expose any vaterite present to dissolution as shells sink through the thermocline into the deep ocean. This may explain the large, enigmatic super-lysocline dissolution of planktic foraminifer shells observed across the

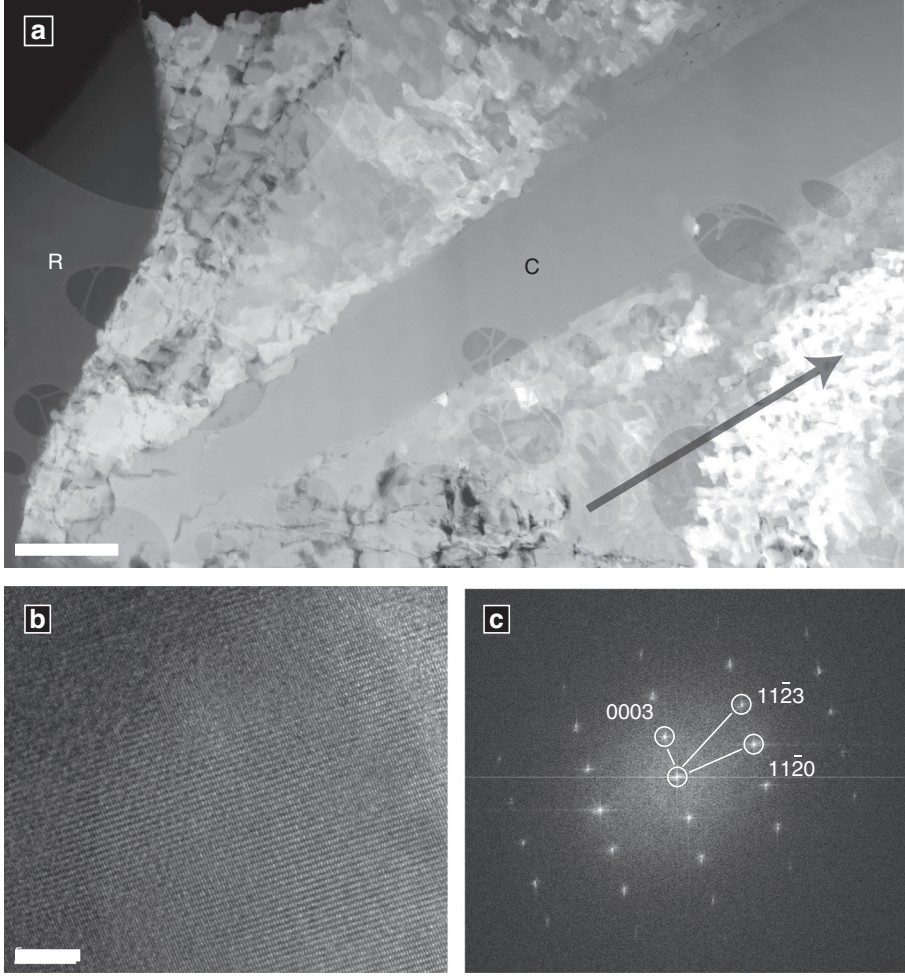

**Fig. 6** Post-transformation structure in *N. dutertrei* shells. **a** Foil #4483 cut from a post-transformation shell displays a large calcite single crystal (C) that disrupts the underlying particulate structure. The dark margin on the left (R) is the sample mounting resin, and the arrow in the lower right points orthogonally towards the inner shell surface. High-resolution TEM (**b**) and indexed Fourier Transform diffraction pattern (**c**) show the crystal is calcite (see also Supplementary Fig. 6). Scale bars 1 μm (**a**), 500 nm (**b**). All analyses carried out in analytical session 2

surface oceans[42]. With increasing ocean acidification, this could reduce the particulate inorganic carbon flux from the surface ocean, with potential significant implications also for any ballasted organic carbon flux to the deep sea and sea floor[43]. A wider survey of the mineralogy of end-of-life foraminifera could evaluate the potential significance of this to the global carbon cycle.

## Methods

**Sample collection and preparation.** Foraminifera were collected from the surface mixed layer and uppermost thermocline (0–50 m depth, 18–20 °C) by plankton tow in the San Pedro Channel (CA, USA) in August 2013. Individual foraminifera were extracted from plankton concentrates using a wide-bore pipette and sacrificed by transfer into ultrapure water for a period of 1–2 h. The shells were then dried at room temperature by placement on absorbent cardboard before being transferred into multi-well micro-palaeontology slides. Large pre-gametogenic adult specimens of *O. universa* (sphere stage) and *N. dutertrei* were selected for further investigation by scanning electron microscopy (SEM) and FIB supported TEM (FIB-TEM) and FTIR Spectrometry (see below).

FIB-TEM analyses were carried out in two separate sessions in March (3 foils from 2 *O. universa*, 1 foil from *N. dutertrei*) and October 2015 (5 foils from 4 *O. universa*, 4 foils from 3 *N. dutertrei*). Samples studied in the first session were lightly crushed under a binocular microscope, fractured pieces were transferred on carbon paste covered SEM stubs and used for FIB milling without further coating. All TEM foils in this session consisted of vaterite with minor ACC. Samples in the second session were from the same batch, but were mounted in epoxy and polished to expose shell cross-sections for FIB milling. All TEM foils analysed in the second session had transformed to calcite.

**Fourier transform infrared spectrometry.** FTIR spectra of single foraminifera shells (two each for *O. universa* and *N. dutertrei*) were measured with a Thermo Nicolet iS10 FTIR spectrometer (Nicolet, MA, USA) equipped with ATR along with a smart performer assessor at Macquarie University. Samples were part of the same batch sampled alive in 2013 and FTIR analyses were carried out in November 2016. Spectra were acquired between 1600 and 500 cm$^{-1}$ with a resolution of 4 cm$^{-1}$ and 64 accumulations. Each analysis was duplicated and four resulting spectra for two shells for each species were averaged and are shown in Fig. 5 and Supplementary Fig. 5. Backgrounds were recorded at the start and the end of the analytical session. Spicules of *H. momus* consisting of stable vaterite[27] and geological calcite from the mineral collection at the Department of Earth and Planetary Sciences, Macquarie University, were measured as comparisons at identical conditions during the same analytical session. Spectra were normalised to the band intensity at 872 cm$^{-1}$ and band assignments were carried out using data from ref. [28] and are tabulated in Supplementary Table 2. Exploiting the linear relationship for relative areas of the bands at 740 and 712 cm$^{-1}$ in vaterite-calcite mixtures[29] the *O. universa* shells measured here contained ca. 4.5% vaterite, while the *N. dutertrei* shells contained ca. 3 % vaterite.

**Focused ion beam sample preparation.** FIB milling using a FEI FIB200 instrument (ex situ lift out method) at the German Research Centre For Geosciences (GFZ) followed procedures published previously[31, 44]. Foils obtained from non-mounted, resin-free sections in the first analytical session transect the entire chamber wall of *N. dutertrei* and the outer half of the ca. 12 μm thick *O. universa* shell. Foils in the second analytical session were cut across resin-mounted sectioned and polished shells.

The FIB-milling method involves sputtering the material surrounding the platinum-protected target area with gallium ions. This process can heat the target area, and drive amorphisation through Ga implantation in the surface of the material[45]. Sample heating is proportional to the beam current, and the extent of

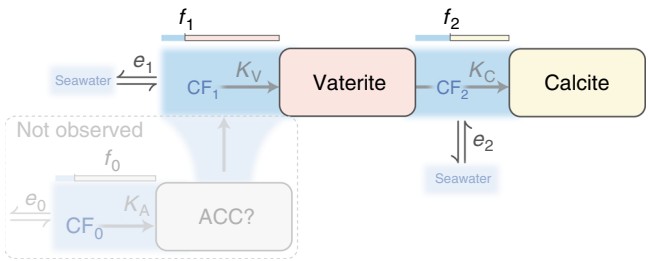

**Fig. 7** The hypothesised double-fractionation mechanism. Initially, vaterite is formed from a calcifying fluid ($CF_1$), which has a starting composition similar to seawater, and exchanges ions with the external environment ($e_1$). The trace element composition of vaterite is determined by vaterite-specific partition coefficients ($K_V$), and the fraction of the available ions precipitated from $CF_1$ ($f_1$), which determines the degree of Rayleigh fractionation during precipitation[48]. Next, vaterite transforms into calcite via a localised dissolution-reprecipitation reaction. This creates a second transient calcifying fluid ($CF_2$) with an initial composition identical to the parent vaterite, which can exchange ions with external fluids ($e_2$, which interact with seawater, $CF_1$ or some other localised reservoir). The composition of the resulting calcite is determined by the composition of $CF_2$, calcite-specific partition coefficients ($K_C$), the degree of Rayleigh fractionation in $CF_2$ ($f_2$) and the degree of ion exchange with external fluids ($e_2$). Conceptually, the trace element content of the mineral gets more similar to CF as $f$ tends towards 1, while $e$ describes how 'open' or 'closed' the CF is. Note that $f$ and $e$ are incompatible: if $e$ is higher than the rate of ion removal by precipitation the system is relatively 'open' and $f$ will be low, and vice versa. Thus, the dynamics of this space depend on the relative rates of crystal precipitation and ion exchange with the external environment. For a double-fractionation scenario to occur, either $e_2$ must be significant (relative to precipitation rate) or $f_2$ must be <100%, otherwise the resulting calcite would inherit the same composition as the parent vaterite. It is further possible that vaterite is preceded by an ACC phase, which would introduce a third fractionation step. Testing this model requires knowledge of element-specific and phase-specific partition coefficients ($^{TE}K_X$), which are not currently available in the literature

amorphisation is proportional to the beam energy, and both depend on the angle of beam incidence during milling[46]. We used 30 keV with a beam current of 11 pA and an angle of incidence of 1.2°. At these conditions beam heating during FIB milling is <10 K[44] and sample amorphisation is minimal. As the foils are thicker than 100 nm, the major part of the foil is thus not affected by ion implantation. If amorphisation were a significant problem in the foils, Debye–Sherrer diffraction rings would be present in all collected diffraction patterns. These features were only observed in diffraction patterns collected at grain boundaries (Fig. 4), which contain clear amorphous regions related to foraminiferal structure.

To date, approximately 5000 FIB foils have been produced at the GFZ TEM facility, of which ca. 1000 foils are of biomineral carbonates and phosphates. Amorphisation introduced by FIB milling across major parts or even the complete thickness of a 100–200 nm thick foil has never been observed. Similarly, transformation of major parts or an entire FIB foil into a different crystalline phase (e.g. calcite to vaterite) using our analytical protocols is considered impossible.

Foils were transferred to individual copper grids coated with holey carbon, before analysis by TEM without further carbon coating.

**Transmission electron microscopy analysis and processing**. A FEI Tecnai™ G2 F20 X-Twin TEM operated at 200 kV acceleration voltage with a field emission gun electron source was used for imaging and analysis, following previously published procedures[31, 44]. Energy-filtered imaging was undertaken with a Gatan Tridiem™ filter applying a 20 eV window to the zero loss peak. Great care was taken to minimise radiation damage to the material during TEM analysis. This involved a low-dose analysis and visual monitoring protocol adapted specifically for the analysis of biominerals and used in previous studies[31]: Foils were analysed in STEM mode, rapidly scanning using a small spot size[8] and assigning the beam between STEM scans to areas outside the sample to avoid electron irradiation damage. At the start of the analytical session for each FIB foil, a rapid overview picture was taken using a defocused beam to identify crystalline and amorphous areas. This overview image was repeated after STEM scanning and HREM analysis and at the end of each analytical session, to confirm that beam damage was not driving sample transformations.

All HR-TEM analyses were carried out at the end of the analytical session for each foil, using exposure times of 0.2 s. For HREM imaging we used a spot size of 5. Diffraction patterns were collected using selected area electron diffraction (SAED), which spreads the beam over a larger region, and exposes the sample to a lower radiation dose than more conventional Convergent Electron Beam Diffraction, at the cost of spatial resolution.

Using this protocol, irradiation damage was only observed on two or three occasions and consisted either of holes from the electron beam or of small amorphisized areas where a STEM scan had been carried out. These areas were discarded from the dataset. In no case was transformation from calcite to vaterite or vice versa observed.

In total, thirty different areas on four foils from *O. universa* and *N. dutertrei* were analysed either by HR-TEM ($n = 17$) or by electron diffraction (SAED) using an image plate detector ($n = 13$). No data filtering was applied except for results shown in Fig. 4c. All diffraction indexing was carried out manually.

**Data availability**. The authors declare that the data supporting the findings of this study are available within the paper and its Supplementary Information files.

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

## Acknowledgements

This work is funded by the ARC via a Future Fellowship to D.E.J. and an ARC Discovery Project to D.E.J., S.M.E. and R.W. The authors are grateful to A. Schreiber for skilful preparation of the FIB foils.

## Author contributions

S.M.E. and D.E.J. designed the experiment, D.E.J., O.B.A.A., R.W. and O.B. carried out the measurements and analysed the data. All authors contributed to the writing of the manuscript.

## Additional information

**Competing interests:** The authors declare no competing financial interests.

