## [Peer Review File · Nature Communications]

Reviewers' comments:

Reviewer #1 (Remarks to the Author):

It has been established since at least 1959 (Blackmon and Todd) that the shells of planktonic, and most other foraminifera are composed of calcite. In this report, Jacob et al studied two species of planktonic foraminifera, and they claim that these species at least are not calcitic, but are composed of vaterite and amorphous calcium carbonate (ACC). This, if correct, would be a seminal contribution. Obviously in order to make the claim that everyone else is wrong, but we are right, the evidence has to be compelling.

The specimens were collected alive in August 2013, they were placed in ultrapure water for 1-2 hrs (ACC readily dissolves in water), dried and then stored until examined some 18 months later in March 2015 and then again in October 2015. The mode of examination was to use the heavy atom beam of the focused ion beam instrument (FIB) to cut very thin foils of shell and then examine these in the TEM both by imaging and electron diffraction. The samples examined in March all showed the presence of vaterite and ACC, whereas the samples examined in October, were all calcite. In other words the results are not reproducible. And this in science is a basic requirement, irrespective of the explanation offered.

The authors explain this disparity by saying that the samples recrystallized between March 2015 and October 2015. I infer that they think that the vaterite and ACC were stable between August 2013, when the samples were collected, and March 2015 ie for about 18 months, and then in the next 6-7 months, they recrystallized. This seems strange and furthermore all samples were vaterite and ACC in March 2015 and then all were calcite in October. I would not expect all the samples to transform in 7 months after 18 months of stability.

The authors recognize that there could be other explanations. One is that the ion beam thinning produced ACC and vaterite – two unstable phases. The argument against this is that the lab that carried out the thinning said that they had produced 4000 foils and none had caused "amorphisation". They don't state whether or not out of these 4000 foils, any were composed of calcium carbonate. They also did not prepare foils from standards of say geological calcite to prove that amorphisation does not occur. This should have been done with each batch of foraminifera samples. Another possibility is that the TEM beam caused radiation damage and this produced the unstable ACC phase and/or the perfectly ordered vaterite crystals. Again no controls and low dose precautions were not taken.

So based on all the above I am unconvinced that Jacob et al have carried out a water tight experiment that proves that planktonic foraminifera, or even these two species, do not produce calcitic shells.

As an aside, I personally have collected living foraminifera (planktonic and benthonic), kept them alive for about 2 hours and then analysed the specimens within minutes using FTIR. The 5 to 10 pooled samples of each species were rinsed in pure water for a few seconds and then dried with acetone, crushed and then the spectrum was obtained immediately. They were all 100% calcite, including *Orbulina*, one of the species examined by Jacob et al. I do not believe that under these conditions all the ACC and vaterite would have recrystallized to calcite. These results were not published for the obvious reason that foraminifera are known to form calcite.

In conclusion, I am not convinced that the conclusions reached by Jacob et al are correct.

Furthermore, I do not think that TEM examination of thin foils is an appropriate method to determine mineralogy especially of unstable phases. FTIR involves drying. What about micro-Raman on living specimens?

Reviewer #2 (Remarks to the Author):

This is an important study that is highly worthy of publication in Nature Communications. The findings stand at the forefront (and intersections) of leading questions in biomineralization, crystal growth and chemical oceanography. As such, it should be considered for publication.

The paper first needs significant editing and development to clearly communicate the mineralogical aspects of the findings and particularly develop the many implications.

The paper would benefit from better documentation of the dogma that assumes forams are making calcite during their lifetime. These results indicate a proper reconstruction would be based on the properties and proxies for ACC and vaterite.

Or perhaps species are specific in making calcite versus ACC and vaterite. And this could be an other explanation for the vital effects that are reported. So much to consider here and all is important.

An important reference to include...

A. Gal et al. *Adv. Func. Material* v 24, 5420-5426

Please see attachment for more comments.

Reviewer #3 (Remarks to the Author):

In their article submitted to *Nature Commun*, Jacob et al. report on the calcified shells of foraminifera, a group of planktic protists. They report for the first time that the shells are constructed from metastable mineral phases of calcium carbonate, i.e. vaterite or even amorphous calcium carbonate. Up to now, it was an unquestioned doctrine that foraminifers are made from direct precipitation of calcite. The finding that metastable phases are involved in shell formation of foraminifers meets well with recent findings in other but multicellular species, and it documents the importance of the importance of nonclassical crystallization processes for in vivo systems. Due to the key role foraminifera take as a geochemical proxy, this finding is of huge importance not only for the biomineralization community but for a wide range of disciplines, especially those which are focussed on (paleo)climate studies.

The provided evidences are conclusive and corroborate well the main claims of the report, i.e. involvement of metastable phases, their transformation and the dissolution behaviour of foraminifers with respect to the calcite lysocline. Nevertheless, I have some points to which I request a comment or correction by the authors and I will detail the points below. However, the points I raise do not affect the major claim of the report. I therefore highly recommend acceptance of this report after minor revisions.

Evidence of Nonclassical Growth in Foraminifers

The authors denote the formation of the foraminifer shell as a nonclassical crystallization process. However, they don't provide direct evidence for a particle-attachment process during growth but infer this from the observation of "pores". The authors interpreted darkish regions occurring throughout the foils as hollow voids which imply an oriented attachment (OA) process. It is fully correct that synthetical mesocrystals formed by OA are typically porous. However, Estroff and others (*Acta Biomater*, 5(8), 3038-44, DOI:10.1016/j.actbio.2009.04.017; *Adv. Funct. Mater.*, 21(11), 2028-2034, DOI:10.1002/adfm.201002709; or *Cryst. Growth Des.*, 12(9), 4574-4579, DOI:10.1021/cg3007734) refer to these voids/pores as intracrystalline organic matrices and also provided evidence for this claim. These "pores" were also observed in sea urchin spines although the spines are fully space-filling and non-porous (*Nanoscale*, 3(2), 603-9, DOI:10.1039/c0nr00697a).

Earlier contributions of Hemleben and Spero have shown a granular organization of the growing shells which support fully the claim of a nonclassical process raised by Jacob et al. If the authors cannot provide any further evidence for a particle-driven growth process (eg. a granular fracture surface or AFM revealing fundamental building blocks, e.g. nanogranules) or that these "pores" are indeed hollow, they should state clearly that their claim of a nonclassical process mainly rests on

earlier reports.

Terminology of Nonclassical Crystallization

With respect to the notion of nonclassical crystallization, the one or other terminological inconsistency can be found in this paper. (i) The authors use multiple time the term of a quasi-single crystal for paraphrasing the notion of a mesocrystal, i.e. a mosaic crystal with a near-to-zero-angle spread. However, the term "quasicrystals" refers to crystals lacking lattice periodicity (e.g. due to unusual symmetry as in Shechtmanite). The IUCr give a definition of quasicrystals in *Acta Crystallographica*, 48(6), 922–946. The most recent definition of a mesocrystal was given by Bergström et al., *Acc. Chem. Res.*, 48, 1391-1402, DOI:10.1021/ar500440b.

L112ff: When your foraminifer shell is consisted of amorphous and therefore isotropic amorphous calcium carbonate, it cannot be formed by oriented attachment since no reference frame is available to which respect the particles can co-align. In this case, should prefer then to terms such as particle attachment, aggregation or accretion.

Apart from these points detailed above, I suggest that the authors should not hide (a) the evidence for ACC in foraminifer shells and (b) the reasoning for the fully justified claim that amorphidization did not occur. Since this data supports strongly the major claim of the paper, I believe the manuscript would be strengthened by incorporating these points in the main body along with Figure S3.

Some minor points:

- 1) Line 29: correct: changes *of* surface
- 2) Line 77: Citation of ref. n°13 (DeYoreo et al in *Science* 2015) is highly advised.
- 3) Figure 1: Labels of the subfigures are probably too small and so is the scale bar in the inset.
- 4) Line 90: Figure 1b shows these perforations, right? If so, refer to this subfigure.
- 5) Line 102: beam => Beam, for consistency.
- 6) For easing the reader's navigation through the supplemental, the ordering of the individual supplemental information should follow the order of occurrence in the main text.
- 7) L126: correct: with and distinct from *those* of calcite.
- 8) L168: Double citation.
- 9) L170/L187: Only few reports give a direct support for ACC-to-crystalline transformation in situ. Another example, which should be cited is *Faraday Disc.*, 159, 433-448.
- 10) L187: Currently, the reports evidencing particle-mediated biominerals growth are very, very scarce. Most of the reports rest on superficial morphology analysis. So far, only one report gives direct in situ evidence of a nonclassical crystallization process in vivo and has to be cited: *Nat Commun*, 6, 10097. DOI: 10.1038/ncomms10097

Answer to reviewer's comments

Reviewer #1 (Remarks to the Author):

It has been established since at least 1959 (Blackmon and Todd) that the shells of planktonic, and most other foraminifera are composed of calcite. In this report, Jacob et al studied two species of planktonic foraminifera, and they claim that these species at least are not calcitic, but are composed of vaterite and amorphous calcium carbonate (ACC). This, if correct, would be a seminal contribution. Obviously in order to make the claim that everyone else is wrong, but we are right, the evidence has to be compelling.

The specimens were collected alive in August 2013, they were placed in ultrapure water for 1-2 hrs (ACC readily dissolves in water), dried and then stored until examined some 18 months later in March 2015 and then again in October 2015.

The mode of examination was to use the heavy atom beam of the focused ion beam instrument (FIB) to cut very thin foils of shell and then examine these in the TEM both by imaging and electron diffraction. The samples examined in March all showed the presence of vaterite and ACC, whereas the samples examined in October, were all calcite. In other words the results are not reproducible. And this in science is a basic requirement, irrespective of the explanation offered.

The authors explain this disparity by saying that the samples recrystallized between March 2015 and October 2015. I infer that they think that the vaterite and ACC were stable between August 2013, when the samples were collected, and March 2015 ie for about 18 months, and then in the next 6-7 months, they recrystallized. This seems strange and furthermore all samples were vaterite and ACC in March 2015 and then all were calcite in October. I would not expect all the samples to transform in 7 months after 18 months of stability.

Answer: While we agree with the reviewer that the disappearance of vaterite over this period is inconvenient, we disagree that it renders our results irreproducible. Vaterite is a highly unstable polymorph, with unstudied transformation pathways. Therefore, the absence of evidence of vaterite in our later samples cannot be taken as evidence of its absence at any previous stage.

We have specifically addressed this concern in our revised manuscript by:

- 1) Including a paragraph (L208-227) explicitly discussing the instability of vaterite, in which we highlight differences in sample preparation processes between the earlier (vaterite) and later (calcite) TEM specimens, which likely drove the phase transformation. The later specimens were mounted in resin and polished before FIB foil extraction, providing a significant energy input and a hydrous environment which could have facilitated the transformation. We further discuss results from a set of wet-stored specimens that were found to be pure calcite after only 3 weeks of storage, which suggest a hydrous environment is important in transformation.
- 2) Included new FTIR analyses of individuals from the same set of samples (L150-158 and Fig. 4), which confirms the presence of vaterite. These analyses were taken *after* the second set of TEM specimens were prepared, indicating that sample preparation procedures are responsible

for vaterite-calcite transformation observed in these specimens. This also provides an independent confirmation of our initial TEM results.

- 3) Added a detailed analytical protocol for sample preparation and TEM for biominerals to the supplementary material.

The authors recognize that there could be other explanations. One is that the ion beam thinning produced ACC and vaterite – two unstable phases. The argument against this is that the lab that carried out the thinning said that they had produced 4000 foils and none had caused “amorphisation”. They don’t state whether or not out of these 4000 foils, any were composed of calcium carbonate. They also did not prepare foils from standards of say geological calcite to prove that amorphisation does not occur. This should have been done with each batch of foraminifera samples. Another possibility is that the TEM beam caused radiation damage and this produced the unstable ACC phase and/or the perfectly ordered vaterite crystals. Again no controls and low dose precautions were not taken.

Answer: We appreciate the reviewer’s concerns about ‘amorphisation’ by FIB techniques. We took great care to evaluate this, given that it is known to create a ~7 nm amorphous damage layer on the outer surface of foraminiferal carbonate wafers (Branson et al, 2015). If this were a significant issue in our samples, we would expect to see a relatively uniform layer of amorphous material across our sample foils, leading to the appearance of Debye-Scherrer rings in all diffraction pattern. We did not see this, which implies that the amorphous damage layer was negligible compared to the crystalline material in the sample.

The amorphous regions we observed in our sample were on the order of a few nm wide, at the margin of the vaterite grains. During FIB preparation, the incident ion beam is always sub-parallel to the milled surface. Alongside the insignificant degree of surface amorphisation across the entire foil, it is extremely unlikely (tending towards physically impossible) that the milling beam would introduce these localized, inter-granular amorphous regions.

However, following Dr. Branson’s recent Atom Probe Tomography study of foraminiferal calcite (Branson et al, 2016), we now note that these localized amorphous regions may be attributable to ACC *or* intra-skeletal organics. Because of the uncertain provenance of these amorphous regions, we have significantly reduced the discussion of ACC in our manuscript.

Finally, the reviewer raises the possibility of transforming calcite to vaterite via FIB milling. This would require the amorphisation of the entire sample wafer, followed by its solid-state transformation back to a highly unstable vaterite phase while maintaining the internal structure and crystallographic orientation of the original sample. We regard this possibility as physically unfeasible, for the following reasons. First, the occurrence of such a transformation in all 30 specimens would imply a highly reproducible transformation. Given that our second set of TEM specimens were prepared in the FIB using identical methods and were all calcite, this is demonstrably not the case. Second, FIB milling has been used to prepare numerous biomineral carbonates (e.g. Jacob et al, 2008, 2011; Branson et al, 2012; Branson et al 2015), and has never been observed to cause such a transformation. Thirdly, significant surface amorphisation would

generate clear Debye-Scherrer rings in all collected diffraction patterns, whereas these features were only observed at inter-grain boundary regions, which contained clear, structure-related amorphous regions. Finally, there is no evidence from any laboratory (peer reviewed or otherwise) that we are aware of suggesting that FIB milling can introduce a systematic phase transformation of this nature. If we are incorrect on this last point, we ask that the reviewer points us towards the relevant reference.

We have addressed the specific concerns regarding FIB sample preparation and amorphisation in the manuscript by:

- 1) Reducing mention of ACC in our manuscript (because we cannot exclude the possibility that it is organic material), and focusing on the primary vaterite result.
- 2) Thoroughly describing FIB processing procedures, quality indicators and the possibility of amorphisation in the supplement.

So based on all the above I am unconvinced that Jacob et al have carried out a water tight experiment that proves that planktonic foraminifera, or even these two species, do not produce calcitic shells.

Answer: We hope that the changes to our manuscript, particularly the addition of new FTIR data, make our case more convincing.

As an aside, I personally have collected living foraminifera (planktonic and benthonic), kept them alive for about 2 hours and then analysed the specimens within minutes using FTIR. The 5 to 10 pooled samples of each species were rinsed in pure water for a few seconds and then dried with acetone, crushed and then the spectrum was obtained immediately. They were all 100% calcite, including *Orbulina*, one of the species examined by Jacob et al. I do not believe that under these conditions all the ACC and vaterite would have recrystallized to calcite. These results were not published for the obvious reason that foraminifera are known to form calcite.

Answer: While the results of the reviewer are indeed interesting, they are difficult to evaluate in light of our work because they are not published. For this reason, we are rather surprised that the reviewer raised them at all, as unpublished data of this nature should not come in to an editor's decision. However, as they have been raised, we are interested to note that the reviewer crushed the foraminifera before analysis. Grinding and crushing are known to induce structural changes, especially when metastable phases are analysed (e.g. Martinez et al., 1981: grinding-induced structural transformations in CaCO_3 , *J. Colloid Interface Sci* 81, 500-510), as was seen in our second set of TEM analyses. This may account for the absence of vaterite in the reviewer's analyses. Our revised manuscript explicitly discusses the issue of instability, and that the ease of vaterite transformation has likely prevented its previous detection in a dedicated paragraph (L208-227).

In conclusion, I am not convinced that the conclusions reached by Jacob et al are correct. Furthermore, I do not think that TEM examination of thin foils is an

appropriate method to determine mineralogy especially of unstable phases. FTIR involves drying. What about micro-Raman on living specimens?

Answer: We hope that the significant changes to our manuscript, and our discussion of the reviewer's main concerns above present a convincing case. In particular, we hope that the addition of FTIR results to independently confirm the presence of vaterite are helpful in this regard.

Regarding the reviewer's aspersions of the FIB-TEM technique, we would argue that, with careful application, the spatial resolution afforded by this technique make it one of the best currently available for analyzing complex biomineral carbonates. This is exemplary in the FIB-TEM work of Dr. Richard Wirth's TEM laboratory at GFZ Potsdam, which has produced numerous published studies of complex biomineral calcium carbonates and calcium phosphates (e.g. Gale et al., 2016, Sviben et al., 2016, Panieri et al., 2017, Jacob et al., 2008, 2011, Lepland et al., 2013), which show excellent preservation of structure and mineralogy, including metastable phases (e.g. Jacob et al., 2011).

Reviewer #2 (Remarks to the Author):

This is an important study that is highly worthy of publication in Nature Communications. The findings stand at the forefront (and intersections) of leading questions in biomineralization, crystal growth and chemical oceanography. As such, it should be considered for publication.

The paper first needs significant editing and development to clearly communicate the mineralogical aspects of the findings and particularly develop the many implications.

The paper would benefit from better documentation of the dogma that assumes forams are making calcite during their lifetime. These results indicate a proper reconstruction would be based on the properties and proxies for ACC and vaterite.

Answer: We have now included a reference to Blackmon and Todd (1959) containing x-ray diffraction data on 131 species of foraminifera.

Or perhaps species are specific in making calcite versus ACC and vaterite. And this could be an other explanation for the vital effects that are reported. So much to consider here and all is important.

Answer: We agree that there are wide-ranging structural and particularly geochemical implications here, particularly regarding the enigmatic 'vital effects' in foraminiferal geochemistry. We have refocused the paper to explicitly consider the geochemical implications of this result, although this discussion is limited by a lack of vaterite geochemical data, and must remain largely speculative.

An important reference to include...

A. Gal et al. Adv. Func. Material v 24, 5420-5426

Answer: Reference included

Please see attachment for more comments.

Answer: We went through the document and included the changes.

Reviewer #3 (Remarks to the Author):

In their article submitted to Nature Commun, Jacob et al. report on the calcified shells of foraminifera, a group of planktic protists. They report for the first time that the shells are constructed from metastable mineral phases of calcium carbonate, i.e. vaterite or even amorphous calcium carbonate. Up to now, it was an unquestioned doctrine that foraminifers are made from direct precipitation of calcite. The finding that metastable phases are involved in shell formation of foraminifers meets well with recent findings in other but multicellular species, and it documents the importance of the importance of nonclassical crystallization processes for in vivo systems. Due to the key role foraminifera take as a geochemical proxy, this finding is of huge importance not only for the biomineralization community but for a wide range of disciplines, especially those which are focussed on (paleo)climate studies.

The provided evidences are conclusive and corroborate well the main claims of the report, i.e. involvement of metastable phases, their transformation and the dissolution behaviour of foraminifers with respect to the calcite lysocline. Nevertheless, I have some points to which I request a comment or correction by the authors and I will detail the points below. However, the points I raise do not affect the major claim of the report. I therefore highly recommend acceptance of this report after minor revisions.

Evidence of Nonclassical Growth in Foraminifers

The authors denote the formation of the foraminifer shell as a nonclassical crystallization process. However, they don't provide direct evidence for a particle-attachment process during growth but infer this from the observation of "pores". The authors interpreted darkish regions occurring throughout the foils as hollow voids which imply an oriented attachment (OA) process. It is fully correct that synthetical mesocrystals formed by OA are typically porous. However, Estroff and others (Acta Biomat, 5(8), 3038-44, DOI:10.1016/j.actbio.2009.04.017; Adv. Funct. Mater., 21(11), 2028-2034, DOI:10.1002/adfm.201002709; or Cryst. Growth Des., 12(9), 4574-4579, DOI:10.1021/cg3007734) refer to these voids/pores as intracrystalline organic matrices and also provided evidence for this claim. These "pores" were also observed in sea urchin spines although the spines are fully space-filling and non-porous (Nanoscale, 3(2), 603-9, DOI:10.1039/c0nr00697a).

Earlier contributions of Hemleben and Spero have shown a granular organization of the growing shells which support fully the claim of a nonclassical process raised by Jacob et al. If the authors cannot provide any further evidence for a particle-driven growth process (eg. a granular fracture surface or AFM revealing fundamental building blocks, e.g. nanogranules) or that these "pores"

are indeed hollow, they should state clearly that their claim of a nonclassical process mainly rests on earlier reports.

Answer: We agree with the reviewer that, while our results are consistent with a particle-attachment process, we do not have direct evidence for it. We have therefore de-emphasised this point throughout our manuscript, and now simply state that the structure we observe is *consistent with* a mesocrystal formed by particle attachment, with reference to other studies (including those mentioned by the reviewer; L113-118).

With regards to the point that Estroff and others report similar ‘pores’ as being intra-crystalline organic matrices, it is not clear whether these organic-filled pores are the primary structure of the organic matrix, or are collected in pores as they are excluded by the growing crystal. In short, our reading of these studies does not exclude a particle attachment process, which would produce organic- or water-filled pores in biominerals (as opposed to hollow ‘voids’ in synthetic mesocrystals). We have noted this in our brief discussion of mesocrystal structure and particle attachment (L113-118)

Terminology of Nonclassical Crystallization

With respect to the notion of nonclassical crystallization, the one or other terminological inconsistency can be found in this paper. (i) The authors use multiple time the term of a quasi-single crystal for paraphrasing the notion of a mesocrystal, i.e. a mosaic crystal with a near-to-zero-angle spread. However, the term “quasicrystals” refers to crystals lacking lattice periodicity (e.g. due to unusual symmetry as in Shechtmanite). The IUCr give a definition of quasicrystals in *Acta Crystallographica*, 48(6), 922–946. The most recent definition of a mesocrystal was given by Bergström et al., *Acc. Chem. Res.*, 48, 1391-1402, DOI:10.1021/ar500440b.

Answer: We agree that this terminology could be confusing, and have changed it accordingly including updating the reference.

L112ff: When your foraminifer shell is consisted of amorphous and therefore isotropic amorphous calcium carbonate, it cannot be formed by oriented attachment since no reference frame is available to which respect the particles can co-align. In this case, should prefer then to terms such as particle attachment, aggregation or accretion.

Answer: Changed

Apart from these points detailed above, I suggest that the authors should not hide (a) the evidence for ACC in foraminifer shells and (b) the reasoning for the fully justified claim that amorphidization did not occur. Since this data supports strongly the major claim of the paper, I believe the manuscript would be strengthened by incorporating these points in the main body along with Figure S3.

Answer: The evidence for inter-granular amorphous material in our specimens is now in the main manuscript. However, following recent observation of organic

structures at similar scale within foraminiferal calcite (Branson et al, 2016), we have de-emphasised its identification as ACC. Rather, we note that the non-crystalline regions could be either ACC or organic material.

Some minor points:

1) Line 29: correct: changes *of* surface

Answer: Abstract is rewritten providing a different perspective

2) Line 77: Citation of ref. n°13 (DeYoreo et al in Science 2015) is highly advised.
Done

3) Figure 1: Labels of the subfigures are probably too small and so is the scale bar in the inset.

Changed

4) Line 90: Figure 1b shows these perforations, right? If so, refer to this subfigure.

Done

5) Line 102: beam => Beam, for consistency. Done

6) For easing the reader's navigation through the supplemental, the ordering of the individual supplemental information should follow the order of occurrence in the main text. Done

7) L126: correct: with and distinct from *those* of calcite. Done

8) L168: Double citation. Removed

9) L170/L187: Only few reports give a direct support for ACC-to-crystalline transformation in situ. Another example, which should be cited is Faraday Disc., 159, 433-448.

Answer: We have re-written this part, which we think now doesn't need this reference to be included.

10) L187: Currently, the reports evidencing particle-mediated biominerals growth are very, very scarce. Most of the reports rest on superficial morphology analysis. So far, only one report gives direct in situ evidence of a nonclassical crystallization process in vivo and has to be cited: Nat Commun, 6, 10097. DOI: 10.1038/ncomms10097

Answer: Reference included. We note that this publication supports the very similar, but earlier findings of Jacob et al. (2008).

References cited here:

Branson, O, Kaczmarek, K., Redfern, S.A.T., Misra, S., Langer, G., *et al.*, (2015) The coordination and distribution of B in foraminiferal calcite. *Earth Planet. Sci. Lett.* **416**, 67-72.

Branson, O., Bonnin, E.A., Perea, D.E., Spero, H.J., Zhu, Z., *et al.*, (2016). Nanometer-scale chemistry of a calcite biomineralization template: Implications for skeletal composition and nucleation. *Proc. National Acad. Sciences USA*, **113**, 12934–12939, doi: 10.1073/pnas.1522864113.

Gal, A., Wirth, R., Kopka, J., Fratzl, P., Faivre, D., Scheffel, A. (2016): Macromolecular recognition directs calcium ions to coccolith mineralization sites. - *Science*, **353**, 6299, 590-593.

Jacob, D. E., Wirth, R., Soldati, A. L., Wehrmeister, U., Schreiber, A. (2011): Amorphous calcium carbonate in the shells of adult *Unionoida*. - *Journal of Structural Biology*, **173**, 2, 241-249.

Jacob, D. E., Soldati, A. L., Wirth, R., Huth, J., Wehrmeister, U., Hofmeister, W. (2008): Nanostructure, composition and mechanisms of bivalve shell growth. - *Geochimica et Cosmochimica Acta*, **72**, 22, 5401-5415.

Lepland, A., Joosu, L., Kirsimäe, K., Prave, A. R., Romashkin, A. E., Črne, A. E., Martin, A. P., Fallick, A. E., Somelar, P., Üpraus, K., Mänd, K., Roberts, N. M. W., van Zuilen, M. A., Wirth, R., Schreiber, A. (2013): Potential influence of sulphur bacteria on Palaeoproterozoic phosphogenesis. - *Nature Geoscience*, **7**, 20-24.

Panieri, G., Lepland, A., Whitehouse, M. J., Wirth, R., Raanes, M. P., James, R. H., Graves, C. A., Crémière, A., Schneider, A. (2017) Diagenetic Mg-calcite overgrowths on foraminiferal tests in the vicinity of methane seeps. *Earth Planet. Scie Lett.* **458**, 203-212.

Sviben, S., Gal, A., Hood, M. A., Bertinetti, L., Politi, Y., Bennet, M., Krishnamoorthy, P., Schertel, A., Wirth, R., Sorrentino, A., Pereiro, E., Faivre, D., Scheffel, A. (2016): A vacuole-like compartment concentrates a disordered calcium phase in a key coccolithophorid alga. - *Nature Communications*, **7**.

Reviewers' comments:

Reviewer #1 (Remarks to the Author):

The new conclusion of the revised version of this paper, namely ““Our HR-TEM findings show that the planktic foraminifer *O. universa* and *N. dutertrei* mineralise their calcite shells via vaterite.” is as pointed out more consistent with many other observations of mineralization processes. As far as I know this is the first time that vaterite is identified as a transient phase. This conclusion is also consistent with the very well documented observation that mature foraminifera shells are composed of calcite.

The additional FTIR data are important. I do note however that the 740cm⁻¹ peak in the forams is not at the same location as the 744cm⁻¹ peak of the standard. This should be commented on as in FTIR these shifts are significant.

I am still confused by the observation that the foils only contain vaterite and some ACC, but not calcite. Maybe I am misunderstanding the text, and if so perhaps the text can be better worded. If not then I am sure readers will also be confused.

Reviewer #2 (Remarks to the Author):

The manuscript presents significant findings that advance our understanding of foram biomineralization. The work is an important step toward improved interpretations of carbonate mineral compositions and is likely to inspire a new way to pose and constrain efforts to interpret paleoenvironmental conditions.

While the paper is vastly improved from the original version, there are significant shortcomings that must be addressed before it can be considered for publication. The authors realize they have interesting observation but it is not yet clearly presented. The organization suggests have not yet developed a clear message.

An extensive rewrite that explains the findings and recognizes the distinction between evidence and the linkages to nonclassical mineralization processes is needed. To allow the imprecisions of the current draft to be published in Science in its current form would be a disservice to the biological, mineralogical, and oceanographic communities.

Broad comments are offered here in the spirit of trying to get this paper where it needs to be but will need more rewriting than a review can provide.

- Please begin by clarifying the methods and results. The opening paragraph refers to living forams. But the data begin by discussing 12 month old stored forams? (line 150-158). Later the discussion works backwards to 6 months, three weeks, etc (lines 217-227). The reader ends completely confused about the initial and subsequent conditions and the evolution of the polymorphs. Storage conditions are added as an additional variable. More confusion and the reader needs to know why storage is important. Consider creating a table that shows age, storage conditions of the forams. Use the table to organize your presentation. Put the table in the Supplemental if that would help. You must be straightforward and clear here. The confusing presentation makes the discussion unclear and feel weasel-ey.

- The emphasis of the paper should be reversed. The central message should be that vaterite is produced by forams as a significant polymorph— not exclusively calcite. The entire community, until now, has based every bit of science on the assumption that calcite is the only component.

This is THE major discovery and is upheld by the data. Finding vaterite intimately intertwined with the calcite (not sure about order of appearance) is quite remarkable and will affect interpretations of biomineral compositions in every imaginable way, including a major step forward to reconciling 'vital effects' as noted by the authors. It is no wonder that most foram calcites show non-sensical calcite signatures. They are probably showing vaterite signatures but no one has considered what those should look like.

- Thus, the paper should focus on the vaterite finding and the many implications. That finding should not be confused with the IMPLICATION of the observations is the realization that metastable phases are involved. This is not the main theme of your paper and the data are not there to prove anything except that it changes during storage (?) If that is your point, please explain the significance of storage to the reader.

- Abstract. It is inappropriate to call out formation via a "nonclassical pathway involving transient metastable phases" based on the data presented in this paper. This is an overstatement in the abstract (lines 31-33). This is an important implication but should not be confused as a finding unless your point is about storage. You can see this is confusing.

- The paragraph (lines 237-260) begins with a thoughtful approach to the overall conceptual model for the observations. Several concepts are mixed together and this could be more clear by creating two focused paragraphs.

- Again, it is critical to clarify the findings and the confusion of the time-dependent observations.

- There is a consensus in the international community that the term 'mesocrystal' was a misleading concept. Helmut Colfen is fond of the word because it was coined in his laboratory, but you may wish to look to the future and consider removing that term from the paper.

- There is also a significant problem with the term 'precursor' which means something entirely different from anything observed or discussed in this paper. As you look to the long-term durability of this work, 'precursor' should be removed from the manuscript. Suggestions for replacement include 'metastable polymorph(s)', thermodynamic intermediate(s).

- Also note, the concept of particle assembly refers to crystal growth in a bulk solution where particle motion and reorientation is possible. As noted in your paper, foram biominerals grow in an organic matrix without 'bulk solution'. Your findings beautifully demonstrate the biomineral is a particle-based material but you can only infer process from the static evidence. Don't overstate the science and lower the credibility of your findings.

- Lines 262-297 speak to the important point that the measured signatures (isotopic and elemental) are indicative of the initial polymorph that formed. That could be ACC but isn't yet documented for this system. If you are going to speculate on double fractionation, then you may wish to note the equally probable (but admittedly not observed) ACC to crystal is probably the earliest step.

- Last paragraph. Make clear that the greater solubility of the vaterite polymorph is the basis for why forams may be more susceptible to an increasingly acidic ocean.

In summary, this paper is greatly improved and has promise to be an important and highly cited paper but needs work.

The presentation of --findings-- must be clearer and separated from implications. What are the biomineral phases in the living foram? The 'stored' foram? The effects of storage? Are most interpretations of forams coming from sediment samples stored in the lab? Is this what is important? Relations to living forams?

Make a list of the implications that you wish to emphasize and clearly present. The discussion indicates a clear message is not yet organized. There are the issues for interpreting foram signatures. there are the connections to the new paradigm that most biominerals are composed of particles and that the particles we see now were transformed from an initial metastable phase? There is the implication for acidified oceans.

Reviewer #3 (Remarks to the Author):

The revised manuscript sufficiently addressed all point I raised and I have no objections against acceptance. Therefore, I fully support and strongly recommend – without any reservation – publication of this manuscript in its current state.

Reviewers' comments and our answers (in blue):

Reviewer #1 (Remarks to the Author):

The new conclusion of the revised version of this paper, namely ““Our HR-TEM findings show that the planktic foraminifer *O. universa* and *N. dutertrei* mineralise their calcite shells via vaterite.” is as pointed out more consistent with many other observations of mineralization processes. As far as I know this is the first time that vaterite is identified as a transient phase. This conclusion is also consistent with the very well documented observation that mature foraminifera shells are composed of calcite.

The additional FTIR data are important. I do note however that the 740cm⁻¹ peak in the forams is not at the same location as the 744cm⁻¹ peak of the standard. This should be commented on as in FTIR these shifts are significant.

The peak shift is within the spectral resolution for these analyses of 4 cm⁻¹; thus it is not significant, as stated in the figure caption and supplementary information. We have modified the figure caption to make this clearer, and mention possible structural variations between the foraminiferal and reference vaterite.

I am still confused by the observation that the foils only contain vaterite and some ACC, but not calcite. Maybe I am misunderstanding the text, and if so perhaps the text can be better worded. If not then I am sure readers will also be confused.

The reviewer is correct in that the TEM foils measured in the first analytical session do not contain calcite. This does not exclude the presence of calcite somewhere else in the shell that was not sampled by the foil (note the foil is 5µm by 15µm and 150nm in thickness, thus only samples a minute area of the outer surface of the shell). To avoid any potential confusion, we have re-organised our results and discussion sections so that all our observations are laid out clearly and chronologically, with explicit links and comparisons drawn between the different analytical sessions.

Reviewer #2 (Remarks to the Author):

The manuscript presents significant findings that advance our understanding of foram biomineralization. The work is an important step toward improved interpretations of carbonate mineral compositions and is likely to inspire a new way to pose and constrain efforts to interpret paleoenvironmental conditions.

While the paper is vastly improved from the original version, there are significant shortcomings that must be addressed before it can be considered for publication. The authors realize they have interesting observation but it is not yet clearly presented. The organization suggests have not yet developed a clear message.

An extensive rewrite that explains the findings and recognizes the distinction between evidence and the linkages to nonclassical mineralization processes is needed. To allow the imprecisions of the current draft to be published in Science in its current form would be a disservice to the biological, mineralogical, and oceanographic communities.

We thank the reviewer for their suggestions to improve our manuscript, and offer specific replies to their comments below. In our overall reading of their comments, it appeared to us that the main problem with our manuscript was the reporting of additional results in the discussion, which had not been previously mentioned in the results. This introduced a complex topic late on in the discussion, and meant that key statements in the early discussion were not directly supported by our results section. As a result, the reviewer correctly noticed that a number of our key stated findings were lacking support, and suggested an extensive re-write in which “the emphasis of the paper should be reversed”.

While we appreciate that our previous draft was significantly lacking, we believe that the misunderstandings leading to the reviewer suggesting that the emphasis of the paper be reversed can be dealt with by improving the organization and structure of our manuscript, rather than with a complete re-write. To address this, we have re-organised our results section to explicitly mention all analyses carried out, and draw links and comparisons between them, providing a firmer basis for our discussion. We hope that this re-organization addresses the reviewer’s concerns.

Broad comments are offered here in the spirit of trying to get this paper where it needs to be but will need more rewriting than a review can provide.

- Please begin by clarifying the methods and results. The opening paragraph refers to living forams. But the data begin by discussing 12 month old stored forams? (line 150-158). Later the discussion works

backwards to 6 months, three weeks, etc (lines 217-227). The reader ends completely confused about the initial and subsequent conditions and the evolution of the polymorphs. Storage conditions are added as an additional variable. More confusion and the reader needs to know why storage is important. Consider creating a table that shows age, storage conditions of the forams. Use the table to organize your presentation. Put the table in the Supplemental if that would help. You must be straightforward and clear here. The confusing presentation makes the discussion unclear and feel weasel-ey.

We agree with the reviewer that how we presented our analyses was inadequate in our previous manuscript, and have carefully rethought their presentation. We have moved all material from the offending paragraph (217-227) in the Discussion to the Results section, and have made the following changes:

- 1) A table containing information on sample storage conditions and length, analytical and sample preparation methods for all four analytical sessions is presented (p. 6).
- 2) For clarity, we explicitly reference these four analytical session throughout results and discussion.
- 3) Results from all analytical sessions are stated in the results section, in chronological order with some brief notes reconciling some of the apparently contradictory findings.

We realise that this last point may be more discursive than is normal for a results section, but feel that this is the clearest and most straightforward way to walk the reader through our analyses, and provide support our later discussion.

- The emphasis of the paper should be reversed. The central message should be that vaterite is produced by forams as a significant polymorph— not exclusively calcite. The entire

community, until now, has based every bit of science on the assumption that calcite is the only component. This is THE major discovery and is upheld by the data. Finding vaterite intimately intertwined with the calcite (not sure about order of appearance) is quite remarkable and will affect interpretations of biomineral compositions in every imaginable way, including a major step forward to reconciling 'vital effects' as noted by the authors. It is no wonder that most foram calcites show non-sensical calcite signatures. They are probably showing vaterite signatures but no one has considered what those should look like.

We agree with the reviewer that the central finding of our study is the presence of vaterite in foraminifera, and this is a core point of the paper. However, as we hope is now apparent from our improved results section, our results also suggest that vaterite is a transient phase in foraminifera, and ultimately transforms to calcite. The exact mechanism and timing (e.g. during life vs after death) of this transition remain uncertain, but it appears to be expedited by hydrous, energetic environments, as evident in the patterns of vaterite presence/absence in our different sample sets.

The reviewer's further comments about the implications of vaterite for our understanding of foraminiferal geochemistry are vital, and are discussed in detail in the final 4 paragraphs of the discussion (line 301 ff).

- Thus, the paper should focus on the vaterite finding and the many implications. That finding should not be confused with the IMPLICATION of the observations is the realization that metastable phases are involved. This is not the main theme of your paper and the data are not

there to prove anything except that it changes during storage (?) If that is your point, please explain the significance of storage to the reader.

We agree that the central finding of our manuscript is the presence of vaterite in foraminifera. However, our results are more nuanced than simply 'foraminifera produce vaterite'. We observe various amounts of vaterite and calcite across numerous analytical sessions and techniques. This provides evidence that foraminiferal vaterite is highly unstable, and likely 'matures' to calcite during the foraminiferal life cycle. This reconciles our results with the widely recognized calcitic mineralogy of foraminifera, and supports the role of vaterite as a transient phase, rather than a mature skeletal mineral. In our previous draft, this information was not presented until mid-way through the discussion, and we therefore understand how our invocation of a non-classical crystallization pathway at the start of the discussion may have seemed like too much of a leap from our relatively straightforward 'presence of vaterite' result. We hope that this inference is better supported by an improved presentation of our rather complex findings in the Results section.

- Abstract. It is inappropriate to call out formation via a "nonclassical pathway involving transient metastable phases" based on the data presented in this paper. This is an overstatement in the abstract (lines 31-33). This is an important implication but should not be confused as a finding unless your point is about storage. You can see this is confusing.

We disagree with the reviewer on this point. We have observed that the shells of living foraminifera consist of vaterite, but it is well known that shells sampled from ocean floor sediments or sediment traps consist of calcite, and we observe a time-dependent transformation of vaterite-calcite between our analytical sessions. The significance of our

variable results across different storage conditions and times imply that transformation from vaterite to calcite proceeds rapidly and easily, and that the 'mature' mineralogy of the foraminiferal shell is calcite, not vaterite. Thus, vaterite is a transient, metastable phase involved in the formation of calcitic foraminifera shells, the presence of which defines a non-classical crystallization pathway.

- The paragraph (lines 237-260) begins with a thoughtful approach to the overall conceptual model for the observations. Several concepts are mixed together and this could be more clear by creating two focused paragraphs.

We have split this paragraph in two, as the reviewer suggests. The first paragraph (lines 269-281) introduces the concept of solid state vs dissolution-precipitation transformation, and the second half identifies a dissolution-precipitation mechanism as the most likely candidate (lines 283-299).

- Again, it is critical to clarify the findings and the confusion of the time-dependent observations.

We hope this has been addressed by our re-organisation of the results section.

- There is a consensus in the international community that the term 'mesocrystal' was a misleading concept. Helmut Colfen is fond of the word because it was coined in his laboratory, but you may wish to look to the future and consider removing that term from the paper.

We agree and have removed this term from the paper. We originally introduced the term 'mesocrystal' following the advice of reviewer #3 in the first round of reviews.

- There is also a significant problem with the term 'precursor' which means something entirely different from anything observed or discussed in this paper. As you look to the long-term durability of this work, 'precursor' should be removed from the manuscript. Suggestions for replacement include 'metastable polymorph(s)', thermodynamic intermediate(s).

We agree and have removed this term from the paper.

- Also note, the concept of particle assembly refers to crystal growth in a bulk solution where particle motion and reorientation is possible. As noted in your paper, foraminiferal biominerals grow in an organic matrix without 'bulk solution'. Your findings beautifully demonstrate the biomineral is a particle-based material but you can only infer process from the static evidence. Don't overstate the science and lower the credibility of your findings.

We agree with the reviewer that it is important to be careful in the interpretation of the findings. In our revised 'ultrastructure' section (line 110 ff) we mention that some of the microstructural features we observe are consistent with those one might expect from colloid attachment and transformation (Gower, 2008; Biomimetic model systems for investigating the amorphous precursor pathway and its role in biomineralization. Chem. Rev. 108, 4551–4627 – references in the revised ms, line 123), a form of particle attachment. We are careful not to invoke the open-solution mechanisms referred to by the reviewer (described in de Yoreo, 2015).

- Lines 262-297 speak to the important point that the measured signatures (isotopic and elemental) are indicative of the initial polymorph that formed. That could be ACC but isn't yet documented for this system. If you are going to speculate on double fractionation, then you may wish to note the equally probable (but admittedly not observed) ACC to crystal is probably the earliest step.

We agree and have added a comment to this (line 321ff.) and in new Fig. 7.

- Last paragraph. Make clear that the greater solubility of the vaterite polymorph is the basis for why forams may be more susceptible to an increasingly acidic ocean.

Done

In summary, this paper is greatly improved and has promise to be an important and highly cited paper but needs work.

The presentation of --findings-- must be clearer and separated from implications. What are the biomineral phases in the living foram? The 'stored' foram? The effects of storage? Are most interpretations of forams coming from sediment samples stored in the lab? Is this what is important? Relations to living forams?

We hope that his point has been addressed by our re-organisation and clarification of the Results section.

Make a list of the implications that you wish to emphasize and clearly present. The discussion indicates a clear message is not yet organized. There are the issues for interpreting foram signatures. there are the connections to the new paradigm that most biominerals are composed of particles and that the particles we see now were transformed from an initial metastable phase? There is the implication for acidified oceans.

While we have not made significant changes to the main message of our manuscript, we hope that our revised structure makes our manuscript clearer. Briefly, our discussion follows the structure:

Paras 1-3: Discussion of the presence of vaterite in foraminifera, its role as a transient metastable phase, and how it might transform to calcite.

Paras 4-6: Implications for foraminiferal geochemistry, including a new, testable multi-step fractionation model in Fig 7.

Para 7: Possible implications for carbonate dissolution in acidified oceans.

Reviewer #3 (Remarks to the Author):

The revised manuscript sufficiently addressed all point I raised and I have no objections against acceptance. Therefore, I fully support and strongly recommend – without any reservation – publication of this manuscript in its current state.

We thank the reviewer very much for this statement.

REVIEWERS' COMMENTS:

Reviewer #2 (Remarks to the Author):

Jacob et al.

The authors have made considerable revisions to the manuscript that have greatly improved the content and presentation. This work should be published after considering the minor comments/corrections below. Each of these is offered in the spirit of further improving the contributions of this work to the broader community.

Six occurrences: the work 'precursor' appears six times in the paper. Please look up the definition of precursor and you will see this is not the term you want to use. Remember this is written for a very broad scientific audience. Better to say vaterite is a metastable phase or reactive intermediate in each of these occurrences.

Line 32. "crystallization pathway involving metastable phases that transform (transient and transform redundant)

Line 36. "(suggestion) Our findings provide a new perspective on how planktic foraminifera form their shells, and may reconcile long-standing inconsistencies in interpretations of geochemical proxies.

The real nugget of the paper seems not quite harnessed in the abstract. Then, it appears in the last two paragraphs of the paper. Can you transmit some of that forward to the abstract? — Geochemical signatures are being captured in vaterite, not calcite. Need to know more about the properties of vaterite (and possibly ACC) as a proxy, especially for those stored samples that appeared to undergo a solid state transformation.

Line 270, 295. Please cite Blue et al., a recent quantitative paper that shows signatures in metastable phases are transmitted to the final calcite products during the transformation. You cannot make direct comparisons to this work but the transformation data support your arguments that current calcite-based proxy models need to be revisited.: Blue, C., A.J. Giuffre, S.T. Mergelsberg, J.J. De Yoreo, and P.M. Dove (2017) Chemical and physical controls on the transformation of amorphous calcium carbonate (ACC) into crystalline CaCO₃ polymorphs. *Geochimica et Cosmochimica Acta*.

Line 247. This sentence no longer says what you intended and uses vaterite three times (and what is 'mature' calcite?) Also, everyone knows vaterite is metastable, so don't accidentally make it sound like you think this is a discovery. Current:saline environment suggests vaterite is a precursor phase that transforms to mature calcite in the natural environment.

Consider something like this as a strawman: "saline environment suggests metastable vaterite is an important early phase to form in the organism that subsequently transforms to stable calcite in the natural environment.

Line 292. Cite Giuffre et al. (currently #32)

Line 321. Bingo paragraph. The opening sentence is closer to the nugget I was looking for in the abstract. Can you place greater emphasis on this in the abstract? Currently seems vague on this key point.

Line 360. Again bingo paragraph. The opening sentence is closer to the nugget I was looking for in the abstract. Can you place greater emphasis on this remark? Currently seems vague on this key point.

This manuscript is an important and timely contribution. It is certain to be widely cited. Thank you for the opportunity to provide comments.

End.

Answer to Reviewers

Reviewer #2 (Remarks to the Author):

The authors have made considerable revisions to the manuscript that have greatly improved the content and presentation. This work should be published after considering the minor comments/corrections below. Each of these is offered in the spirit of further improving the contributions of this work to the broader community.

Six occurrences: the work 'precursor' appears six times in the paper. Please look up the definition of precursor and you will see this is not the term you want to use. Remember this is written for a very broad scientific audience. Better to say vaterite is a metastable phase or reactive intermediate in each of these occurrences.

The Merriam Webster dictionary gives the definition of 'precursor' as: 'a substance, cell, or cellular component from which another substance, cell, or cellular component is formed'. We adopted the reviewer's suggestion and replaced the term in the ms. However, the term is commonly used in the community, which is reflected by the fact that out of the six occurrences in the paper counted by the reviewer, four are in fact in the reference list.

Line 32. "crystallization pathway involving metastable phases that transform (transient and transform redundant)
Changed.

Line 36. "(suggestion) Our findings provide a new perspective on how planktic foraminifera form their shells, and may reconcile long-standing inconsistencies in interpretations of geochemical proxies.
Suggestion accepted.

The real nugget of the paper seems not quite harnessed in the abstract. Then, it appears in the last two paragraphs of the paper. Can you transmit some of that forward to the abstract? — Geochemical signatures are being captured in vaterite, not calcite. Need to know more about the properties of vaterite (and possibly ACC) as a proxy, especially for those stored samples that appeared to undergo a solid state transformation.
Abstract amended.

Line 270, 295. Please cite Blue et al., a recent quantitative paper that shows signatures in metastable phases are transmitted to the final calcite products during the transformation. You cannot make direct comparisons to this work but the transformation data support your arguments that current calcite-based proxy models need to be revisited.: Blue, C., A.J. Giuffre, S.T. Mergelsberg, J.J. De Yoreo, and P.M. Dove (2017) Chemical and physical controls on the transformation of amorphous calcium carbonate (ACC) into crystalline CaCO₃ polymorphs. *Geochimica et Cosmochimica Acta*.
Done

Line 247. This sentence no longer says what you intended and uses vaterite three times (and what is 'mature' calcite?) Also, everyone knows vaterite is metastable, so don't accidentally

make it sound like you think this is a discovery. Current:saline environment suggests vaterite is a precursor phase that transforms to mature calcite in the natural environment. Consider something like this as a strawman: “saline environment suggests metastable vaterite is an important early phase to form in the organism that subsequently transforms to stable calcite in the natural environment.

Suggestion accepted, sentence replaced and repetition of ‘vaterite’ fixed.

Line 292. Cite Giuffre et al. (currently #32)

Done

Line 321. Bingo paragraph. The opening sentence is closer to the nugget I was looking for in the abstract. Can you place greater emphasis on this in the abstract? Currently seems vague on this key point.

Abstract changed accordingly.

Line 360. Again, bingo paragraph. The opening sentence is closer to the nugget I was looking for in the abstract. Can you place greater emphasis on this remark? Currently seems vague on this key point.

Abstract changed accordingly.

This manuscript is an important and timely contribution. It is certain to be widely cited. Thank you for the opportunity to provide comments.

Thank you for a very constructive and respectful reviewing process. Much appreciated.

End.